# Prevalence of bacteriologically confirmed pulmonary tuberculosis and associated risk factors: A community survey in Thirvallur District, south India

Chandra Kumar Dolla[1]*, Bhaskaran Dhanaraj[2], Padmapriyadarsini Chandrasekaran[2], Sriram Selvaraj[1], Pradeep Aravindan Menon[1], Kannan Thiruvengadam[3], Rajendran Krishnan[3], Rajesh Mondal[4], Muniyandi Malaisamy[5], Srinivasa B. Marinaik[1], Lakshmi Murali[6], Srikanth Prasad Tripathy[7]

1 Department of Epidemiology, ICMR-National Institute for Research in Tuberculosis, Chennai, India, 2 Department of Clinical Research, ICMR-National Institute for Research in Tuberculosis, Chennai, India, 3 Department of Statistics, Epidemiology Unit, ICMR-National Institute for Research in Tuberculosis, Chennai, India, 4 Department of Bacteriology, ICMR-National Institute for Research in Tuberculosis, Chennai, India, 5 Department of Health Economics, ICMR-National Institute for Research in Tuberculosis, Chennai, India, 6 District Tuberculosis Centre, MoHFW, Government of Tamil Nadu, Thiruvallur, Chennai, India, 7 ICMR-National Institute for Research in Tuberculosis, Chennai, India

* chandrakumar.d@nirt.res.in

**Data Availability Statement:** All relevant data are within the paper.

## Abstract

### Background

Tuberculosis (TB) prevalence surveys add to the active case detection in the community level burden of TB both national and regional levels. The aim of this study was to assess the prevalence of bacteriologically confirmed pulmonary tuberculosis (PTB) in the community.

### Methods

Household community-based tuberculosis disease survey was conducted targeting 69054 population from 43 villages of 5 blocks in Tiruvallure district adopting cluster sampling methodology of ≥15 years old adult rural population of South India during 2015–2018. All eligible individuals with suspected symptoms of PTB were screened with chest X-ray. Two sputum specimens (one spot and the other early morning sample) were collected for M.tb smear and culture examination. Conversely demographical, smoking and alcohol drinking habits information were also collected to explore the risk factor. Stepwise logistic regression was employed to associate risk factors for PTB.

### Results

A total of 62494 were screened among 69054 eligible population, of whom 6340 were eligible for sputum specimen collection. Sputum for M.tb smear and culture examination were collected in 93% of participants. The derived prevalence of PTB was 307/100000 population (smear-positive 130; culture positive 277). As expected that PTB has decreased substantially compared to preceding surveys and it showed that older age, male, low BMI, diabetes,

**Funding:** This study was supported by the World Health Organization as part of the MODEL DOTS project in the form of funding awarded to CKD and the Indian Council of Medical Research in the form of intramural funding awarded to CKD. The funders had no role in study design, data collection and analysis, decision to publish, or preparation of the manuscript.

**Competing interests:** The authors have declared that no competing interests exist.

earlier history of TB and alcohol users were significantly associated (p < .0001) with an increased risk of developing PTB.

## Conclusion

Upshot of the active survey has established a reduction in the prevalence of PTB in the rural area which can be accredited to better programmatic implementation and success of the National TB Control Programme in this district. It also has highlighted the need for risk reduction interventions accelerate faster elimination of TB.

## Introduction

The National Strategic Plan (NSP) for TB 2017–2025 prepared by the Ministry of Health and Family Welfare, Government of India in consultation with 150 national and international public health experts, program managers, donor agencies, technical partners, civil societies, community representatives and other stakeholders in TB control both from the public as well as the private sector [1]. One of the goals of NSP 2017–2025 was to conduct the TB prevalence surveys to monitor the National TB Elimination Programme. There is an over-all epidemiological dissimilarity in TB prevalence between urban, and rural areas. While the urban TB prevalence surveys are characterized by an inferior TB prevalence with moderate annual risk, while rural areas revealed higher TB prevalence with lower annual risk [1] The diversity of TB epidemiology in the country necessitates a periodic measure of the burden of TB for addressing the problem. Tuberculosis burden estimate expert Peter J Dodd expressed the importance of disease burden estimates, which would suggest advocacy, scheduling, ranking, development assistance funding applications and allocating resources to monitor progress at the global level towards End TB strategy [2].

Since 1968, the world recognized 15-year old Bacillus Chalmette Guerin (BCG) vaccine trial was conducted by ICMR-National Institute for Research in Tuberculosis (NIRT), Chennai in the Thiruvallur district of Tamil Nadu, south India [3,4]. This population was monitored intensively by subsequent active community surveys over a period of fifty years, which evaluated the disease burden of tuberculosis—where the present survey was carried out to estimate the PTB burden. The survey followed methodology of preceding surveys and the statistical procedure suggested by WHO guideline [5].

## Methodology

### Study area

The National Institute for Research in Tuberculosis, one of the ICMR institutes, conducted a series of TB prevalence surveys in Thiruvallur district in the southern part of India since 1968. Thiruvallur is located about 50 kilometres from Chennai and includes five administrative blocks with 3,728,104 population. It has 17 Peripheral Health Institutions (PHIs) and two government general hospitals. The scientifically established the community-based Tuberculosis (TB) prevalence survey commenced on July 15, 2015, to study the geographical distribution of tuberculosis cases in it. The required sample population was selected with prior prevalence (3/1000) and adopting cluster randomization through probability proportional sampling design including block design concept. Census commenced targeting 69054 population from 47 villages of 5 blocks at Thiruvallure district. Apart from this, the household contacts aged 15 years

and below of confirmed TB cases in the family, were included in this survey. The eligible population was aged ≥15 years, both males and females from the rural villages were contacted at their residence by the study team and the purpose of the screening was explained to them in the local language. An informed written consent was obtained from the participant after ascertaining their willingness.

## Sampling design and sample size

Community-based cross-sectional study was planned in the villages/urban areas in the five blocks of Thiruvallur district. A multistage stratified cluster sampling design was adopted to select the clusters. The study area was stratified according to the blocks and the total sample size was proportionally allocated to the blocks. The clusters here were the villages/wards in rural/urban areas respectively. The sample size was proportionally allocated to 5 blocks and the clusters within the blocks were selected by a simple random sampling method. In the selected cluster, all the households were covered.

The earlier prevalence surveys in the study area (from 1999 to 2008) showed a declining trend in the culture positive TB prevalence with an annual reduction rate of 5.8%. Assuming the same annual reduction rate, it was estimated that in 2015, the culture positive TB prevalence will be 255 per 100,000 population. Assuming this prevalence rate, a precision of 20%, with a 95% confidence level, design effect of 2 and coverage of 90%, the sample size was estimated to be 83,155 which is approximately 84,000. Assuming the same annual reduction rate and based on the prevalence trend in the survey area, it was estimated that a population of 100867 was needed for this community survey to achieve the adult population.

## Community preparedness

Before the survey was initiated, the researcher did k community preparedness. The study team including supervisor, Doctors, Statisticians, field supervisors, field investigators visited each selected village and met village leaders and explained about the study purpose and procedures to assist our survey team and extend hassle-free coordination between local villagers and research team. The village maps were collected and the sketch of the streets and lanes were prepared for census operation. The entire team was divided into four groups i.e. census enumeration, screening, sputum team and x-ray teams.

## Survey procedures

Survey procedures were carried out according to standard operating procedures (SOP) followed by the previous surveys by a trained team of enumerators, symptom elicitors, X-ray technicians, sputum collectors, planners, and supervisors. All members of the team were well trained, experience ranging about 15 years to 30 years in the field in the past TB prevalence survey. All the staff involved in this study were trained in survey methods uniformly as per the current protocol and implemented. A team member interviewed every member of the house about their health status relevant to TB related symptoms like cough that lasts 3 weeks or longer, pain in the chest for ≥1 month, coughing up blood or sputum, weakness or fatigue, weight loss, no appetite, chills, unexplained fever for ≥1 month and sweating at night. Also information of demography, smoking and drinking habits were collected and fed in the server using a pre-formatted android Personal Digital Assistant (PDA) to explore the risk factor. The team member sent every adult of the household to mobile X-ray car for X-ray screening and made a notecard to sputum team for sputum collection if they had any clinical symptom on the spot. The laboratory procedures and reading of digital X-rays were performed by experienced qualified doctors of NIRT.

## Enumeration and registration

The prearranged feasibility to conduct a survey in the community during early morning and late evenings hours of a day, in order to meet all the household members to maximise the coverage of the targeted population. The survey team went to every household and started with marking household number, which was unique to this survey for future reference. Each enrolled individual was assigned a unique identification (ID) number and directed registration to screen the participants according to the presence of on spot clinical symptoms of TB for sputum collection.

## Digital X-ray screening for TB

The X-ray team contented three members' one X-ray technician, one well-trained laptop operator where the digital X-ray directly displayed from the X-rays machine and one patient handler. The X-ray technician was trained to follow the standardized procedure of X-ray handling to the individual subjects. All eligible household members were directed by enumerators to X-ray car after notifying individual unique ID on the small white card to digital X-ray operator. Each registered individual was screened by a chest radiograph (digital X-ray) in the village through Mobile digital X-ray unit for TB after obtaining informed written consent for adult and assent for those less than 15 years. The radiographs were read independently by two readers and in case of disagreement, by a third reader. All individuals interviewed by trained field investigators for the presence of one or more of the symptoms suggestive of TB. History of anti-TB treatment previous/current anti-TB treatment was also elicited from each individual. For those with an abnormal chest radiograph and/or chest symptoms or previously diagnosed cases in the earlier surveys, two sputum samples (one spot and one early morning) were collected.

## Sputum collection team

As directed by enumerator team, the patients those who had on spot clinical symptom and X-ray abnormality, the sputum team collected two sputa both in the early morning and on the spot sputum and stored in the sputum storage kit. The samples were sent to Laboratory while the related information card was sent to Epidemiology Statistical division for further coordination among laboratory and data entry process in Electronic Data Processing unit (EDP). Furthermore, this team also ensured patients on treatment and provided counseling to continue the treatment. All households in the entire cluster were systematically covered.

## Laboratory investigations

Both the collected sputum along with clinical symptom and abnormal X-ray of each individual, were sent to the laboratory with sputum storage kit by sputum team. The sputum specimens were processed for smear fluorescent microscopy using auramine O stain, culture with solid culture using Lowenstein Jensen (L-J) medium. Samples were inoculated on slopes (two) of L-J medium and monitored every week for a period of eight weeks for growth. When there was no growth after eight weeks the culture was reported negative. Drug Susceptibility Testing (DST) was done for fist line anti-tuberculosis drugs (streptomycin, isoniazid, rifampicin and ethambutol) using Mycobacterium Growth Indicator Tube (MGIT).

## Data management

The data collection and data management were adopted in this survey through Personal digital assistant (PDA), which was developed such way it could be used even in the interior rural

villages where no internet facility was not available and implemented. The data were being collected in offline and syncing in the field station at the point of internet facility to NIRT's central server, which could be viewed in the same day of data collected on a real-time database. The PDA application was developed by outsourcing, which contents of four modules i.e. Census taker, field investigator, X-ray taker, sputum collector and administrator modules. All the module users were given separate IDs and Password to collect independently. The interview schedule was installed in PDA and administered to collect information on demographic characteristics (age, sex), lifestyle characteristics (tobacco smoking, alcohol use), biomass fuel usage for cooking, clinical characteristics such as BCG scar, history of TB, and chest symptoms suggestive of TB. Further details for body mass index calculation; height in centimeters, weight in kilograms were measured. The hypertension was measured by a digital blood pressure monitor, diabetes by drug treatment history to all participants.

## Definitions used

**Smear positive TB.**   If the smear of either of the sputum specimen showed one or more acid-fast bacilli (AFB), irrespective of culture result.

**Culture positive TB.**   If either of the specimens exhibited one or more colonies of M. tuberculosis, irrespective of smear result.

**Bacteriologically positive TB.**   If at least one of the two specimens was a smear and/or culture positive.

## Quality control

A quality check on symptom screening was done by a supervisor on a random sample of 10% of subjects. This was done independently without seeing the original card.

## Statistical analysis

The data were collected through PDA and synced into NIRT's central server, which was being monitored by Epidemiology Statistical unit on a real-time basis and immediate error verification after downloading all the modules data and merged and checked for completeness, errors by the statistical team on a daily basis. The error list was sent if any to the field unit for immediate correction, which found only a 1% error. Finally, the master data set was locked for analysis. The data were cleaned and checked missing information by using the checklist of the survey. The estimated prevalence of TB was stratified by smear-positive and culture positive. Initiated with data-driven methodology engaging each variable checking with variation, out layer and distributional assumption. Inception with univariate analysis and finally selected variable for multivariate technic stepwise logistic regression was employed for age, gender, BMI, diabetes, hypertension, history of TB tobacco smoking, alcohol consumption and cooking fuel used in the household, which was recorded as a categorical variable to engage the analysis by using SPSS 25.0 software (SPSS Inc., Chicago, IL, USA).

## Ethical considerations

This study protocol was approved by the Institutional Ethics Committee of the National Institute for Research in Tuberculosis, Indian Council of Medical Research, Chennai (IEC No: NIRT-IEC-2014032). All diagnosed TB patients were referred to the nearest NTEP for further management as per the guidelines.

## Results

### Coverage for screening

A total of 88080 population registered by the census for this survey. Out of these 69054 were met the eligibility criteria for screening TB. Among the eligible 90.5% (N = 62494) covered for screening (Fig 1). Of 69054 made attempt to screen, 6560 (9.5%) either refused or were not available during the survey.

### Rationale for achieved power

The study power was achieved earlier part of this survey even though the planned 84000 samples were to be achieved but the study was stopped at this stage of completion 62494 were

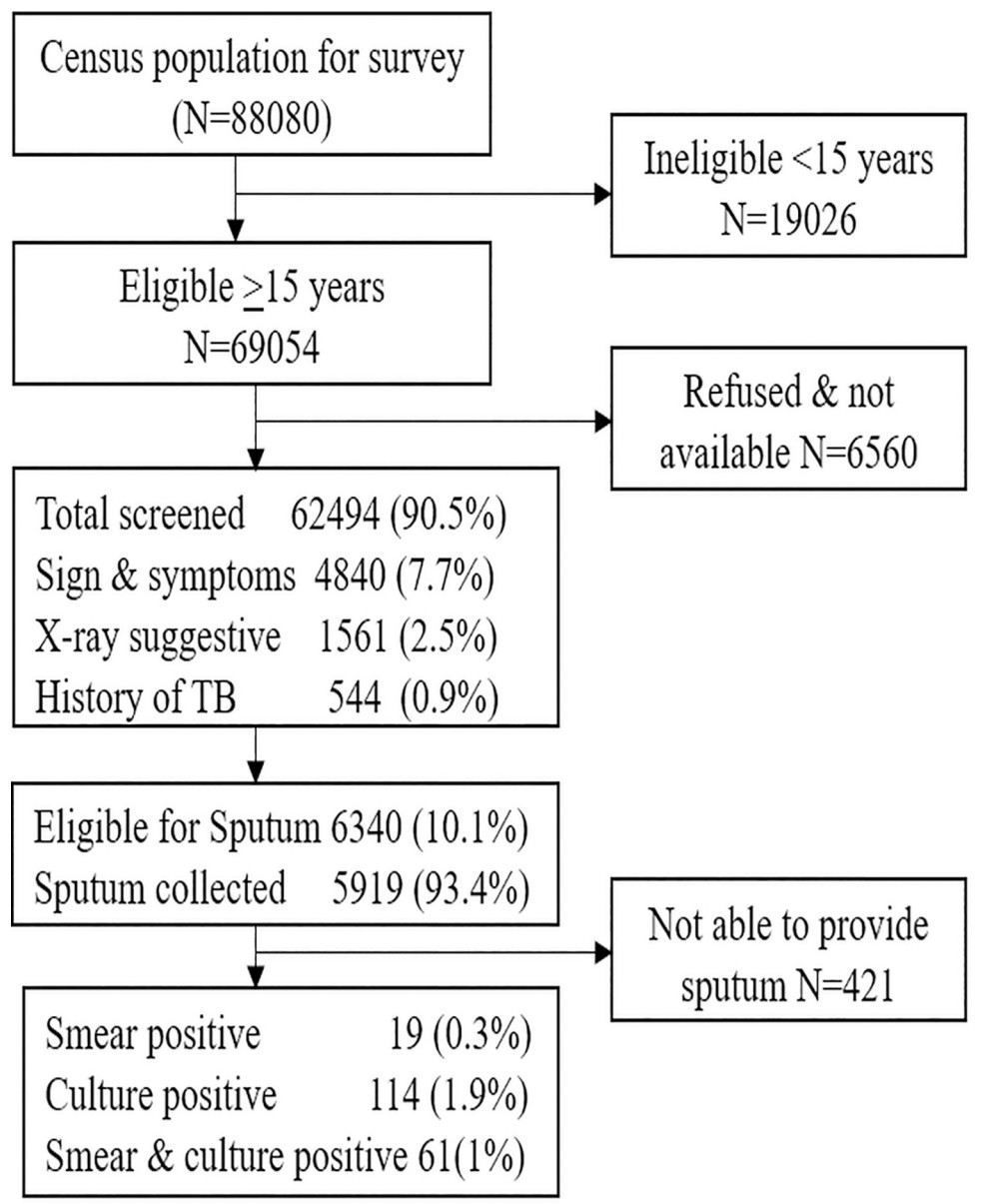

**Fig 1. Study population and its coverage.**

screened among 69054 eligible population from 43 clusters out of 47. The precision was comparatively high because we estimated power on the completed sample size, the type II error was minimum. In this case, the sample size was lesser than the estimated sample size. In this study, if the remaining 4 clusters were not surveyed the demographic prevalence of the disease or the information regarding the demographic prevalence was lacking even though the estimated power has been achieved in this completed population. The prevalence has been estimated on the planned sample size by the achieved population, which is scientifically rationale for comparison. The completed 62494 screened sample has been considered for analysis and estimation.

## Prevalence of TB suspects

A total of 62494 screened, the prevalence of chest symptomatic suggestive TB was 7.7% (n = 4840), X-ray abnormality suggestive of TB was 2.5% (N = 1561) and persons with a previous history of TB was 0.9% (N = 544). We found that the overall prevalence of TB suspects in this population was 10.1% (N = 6340). It was observed that the prevalence of TB suspects was higher among the elderly population (X-ray abnormality was 7.1%; chest symptomatic 13.3%). With respect to gender, X-ray abnormality was almost similar (2.1% vs 2.9%) whereas chest symptoms (5.9% vs 9.8%) and history of TB (0.6% vs 1.2%) were higher in males as compared to females (Table 1).

## Coverage for laboratory investigations

The proportion eligible for laboratory examination was 6340, among them 93.4% (N = 5919) sputum samples were collected. The coverage was similar in all age groups and in both genders (Table 1). Remaining 6.6% (N = 421) not able to collect sputum due to their inability to provide.

## Overall prevalence of bacteriologically confirmed PTB

The total number of pulmonary tuberculosis cases detected in this survey was 192 (Table 2). The estimated prevalence of bacteriologically positive PTB was 307 per 100000 population. The prevalence of PTB increases as age increases (46 to 721 for 15–34 to above 55 years respectively). It was observed that a significantly higher prevalence of PTB among males than the females (85 vs 555). The overall prevalence of PTB was significantly high among persons with

**Table 1. Study population and coverage for the prevalence survey.**

| | Eligible population | Population screened | X-ray suggestive of TB | | Chest symptomatic | | History of TB | | Eligible for sputum collection | | Sputum collected | |
|---|---|---|---|---|---|---|---|---|---|---|---|---|
| | No | No | No | % | No | % | No | % | No | % | No | % |
| **Age Groups** | | | | | | | | | | | | |
| 15–34 | 29105 | 25938 | 116 | 0.4 | 1182 | 4.6 | 113 | 0.4 | 1383 | 5.3 | 1226 | 88.6 |
| 35–54 | 25251 | 22824 | 476 | 2.1 | 1825 | 8.0 | 242 | 1.1 | 2359 | 10.3 | 2183 | 92.5 |
| ≥ 55 | 14698 | 13732 | 969 | 7.1 | 1833 | 13.3 | 189 | 1.4 | 2598 | 18.9 | 2510 | 96.6 |
| **Gender** | | | | | | | | | | | | |
| Female | 35296 | 32956 | 692 | 2.1 | 1954 | 5.9 | 200 | 0.6 | 2658 | 8.1 | 2457 | 92.4 |
| Male | 33758 | 29538 | 869 | 2.9 | 2886 | 9.8 | 344 | 1.2 | 3682 | 12.5 | 3462 | 94.0 |
| **Total** | 69054 | 62494 | 1561 | 2.5 | 4840 | 7.7 | 544 | 0.9 | 6340 | 10.1 | 5919 | 93.4 |

**Table 2. Estimated prevalence of TB and its associated risk factors.**

| | Population | Smear positive TB | Culture positive TB | Bacteriologically positive TB | Prevalence of TB per 100000 pop. | | | OR (95% CI) | p-value | aOR (95% CI) | p-value |
|---|---|---|---|---|---|---|---|---|---|---|---|
| | | | | | Smear positive | Culture positive | Bacteriologically positive | | | | |
| **Age** | | | | | | | | | | | |
| 15–34 | 25938 | 4 | 11 | 12 | 15 | 42 | 46 | 1 | | 1 | |
| 35–54 | 22824 | 43 | 73 | 81 | 188 | 320 | 355 | 7.79 (3.91–15.16) | <0.001 | 8.92 (4.37–18.19) | <0.001 |
| ≥ 55 | 13732 | 34 | 89 | 99 | 248 | 648 | 721 | 15.69 (7.93–31.06) | <0.001 | 13.54 (6.66–27.51) | <0.001 |
| **Gender** | | | | | | | | | | | |
| Female | 32956 | 10 | 26 | 28 | 30 | 79 | 85 | 1 | | 1 | |
| Male | 29538 | 71 | 147 | 164 | 240 | 498 | 555 | 6.57 (4.11–10.50) | <0.001 | 4.73 (2.91–7.68) | <0.001 |
| **BCG Scar** | | | | | | | | | | | |
| No | 37375 | 51 | 107 | 117 | 136 | 286 | 313 | 1 | | | |
| Yes | 25119 | 30 | 66 | 75 | 119 | 263 | 299 | 0.95 (0.67–1.36) | 0.792 | | |
| **BMI** | | | | | | | | | | | |
| ≥18.5 | 52627 | 45 | 91 | 104 | 86 | 173 | 198 | 1 | | 1 | |
| <18.5 | 9867 | 36 | 82 | 88 | 365 | 831 | 892 | 4.55 (3.40–6.07) | <0.001 | 4.48 (3.26–6.15) | <0.001 |
| **Diabetes** | | | | | | | | | | | |
| No | 59818 | 73 | 157 | 176 | 122 | 262 | 294 | 1 | | 1 | |
| Yes | 2676 | 8 | 16 | 16 | 299 | 598 | 598 | 2.04 (1.17–3.56) | 0.012 | 1.67 (0.99–2.83) | 0.055 |
| **Hypertension** | | | | | | | | | | | |
| No | 59561 | 76 | 166 | 184 | 128 | 279 | 309 | 1 | | | |
| Yes | 2933 | 5 | 7 | 8 | 170 | 239 | 273 | 0.88 (0.46–1.71) | 0.711 | | |
| **History of TB** | | | | | | | | | | | |
| No | 61522 | 64 | 151 | 165 | 104 | 245 | 268 | 1 | | 1 | |
| Yes | 972 | 17 | 22 | 27 | 1749 | 2263 | 2778 | 10.62 (6.55–17.24) | <0.001 | 3.64 (2.20–6.02) | <0.001 |
| **Tobacco smoking** | | | | | | | | | | | |
| Never | 56573 | 40 | 97 | 107 | 71 | 171 | 189 | 1 | | 1 | |
| Current | 4848 | 36 | 66 | 75 | 743 | 1361 | 1547 | 8.29 (5.87–11.71) | <0.001 | 1.26 (0.83–1.93) | 0.281 |
| Past | 1073 | 5 | 10 | 10 | 466 | 932 | 932 | 4.96 (2.78–8.86) | <0.001 | | |
| **No of cigarettes/Beedies per day** | | | | | | | | | | | |
| Nil | 56573 | 40 | 97 | 107 | 71 | 171 | 189 | 1 | | | |

*(Continued)*

**Table 2.** (Continued)

| | Population | Smear positive TB | Culture positive TB | Bacteriologically positive TB | Prevalence of TB per 100000 pop. | | | OR (95% CI) | p-value | aOR (95% CI) | p-value |
|---|---|---|---|---|---|---|---|---|---|---|---|
| | | | | | Smear positive | Culture positive | Bacteriologically positive | | | | |
| 1_10 | 4372 | 25 | 46 | 52 | 572 | 1052 | 1189 | 6.35 (4.64–8.69) | <0.001 | | |
| 11_20 | 792 | 8 | 11 | 12 | 1010 | 1389 | 1515 | 8.12 (4.12–16.01) | <0.001 | | |
| >20 | 757 | 8 | 19 | 21 | 1057 | 2510 | 2774 | 15.06 (8.12–27.91) | <0.001 | | |
| **Alcohol consumption** | | | | | | | | | | | |
| No | 52366 | 36 | 93 | 104 | 69 | 178 | 199 | 1 | | 1 | |
| Current | 9115 | 40 | 66 | 74 | 439 | 724 | 812 | 4.11 (2.88–5.88) | <0.001 | 1.62 (1.14–2.32) | 0.007 |
| Yes | 1013 | 5 | 14 | 14 | 494 | 1382 | 1382 | 7.04 (3.66–13.56) | <0.001 | | |
| **Alcohol consumed (ml)** | | | | | | | | | | | |
| Nil | 52366 | 52 | 126 | 139 | 99 | 241 | 265 | 1 | | | |
| <200ml | 10128 | 29 | 47 | 53 | 286 | 464 | 523 | 3.02 (2.06–4.43) | <0.001 | | |
| **Type of cooking fuel** | | | | | | | | | | | |
| Not Applicable | 31020 | 18 | 50 | 54 | 58 | 161 | 174 | 1 | | | |
| Smokeless | 16254 | 39 | 78 | 90 | 240 | 480 | 554 | 3.19 (2.29–4.46) | <0.001 | | |
| Smoke | 15220 | 24 | 45 | 48 | 158 | 296 | 315 | 1.81 (1.04–3.18) | 0.037 | | |
| **Overall** | **62494** | **81** | **173** | **192** | **130** | **277** | **307** | | | | |

low BMI (892), persons with a previous history of TB (2778), diabetes (598), current smokers (1547), the intensity of smoke >20 cigarettes per day (2447) and alcohol users (1382).

## Prevalence of smear-positive PTB

The prevalence of smear-positive PTB was 130 cases per 100000 population (Table 2). It was significantly high among persons aged ≥55, males, low BMI, diabetes, persons with the previous history of TB, current smokers, those who smoked >20 cigarettes per day and alcohol users (1382).

## Prevalence of culture positive PTB

The prevalence of culture positive PTB was 277 cases per 100000 population (Table 2). It was significantly high among persons aged ≥55, males, low BMI, diabetes, persons with a previous history of TB, current smokers, those who smoked >20 cigarettes per day and alcohol users (1382).

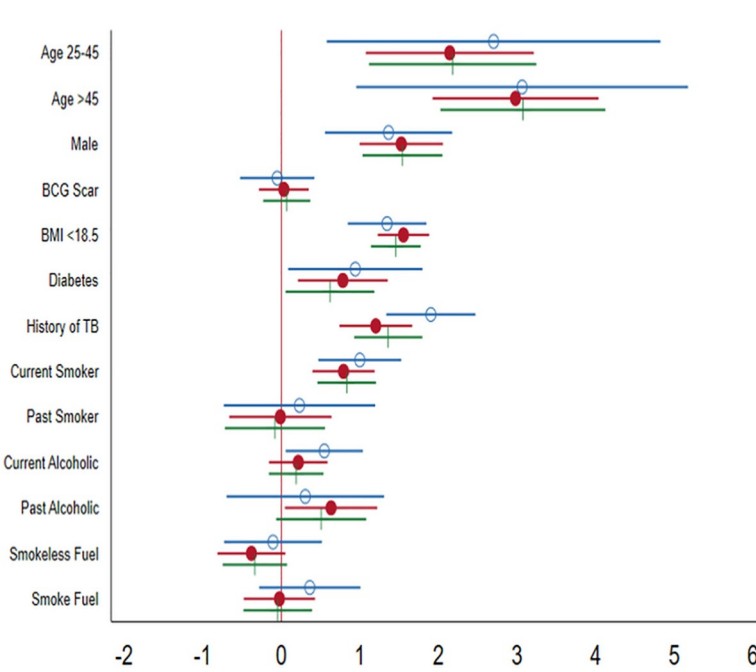

**Fig 2. Risk factors associated with TB.**

### Risk factors associated with bacteriologically PTB

Univariate analysis identified a wide range of factors associated with an increased risk of PTB (Table 2). Stepwise logistic regression analysis found that increasing age (aOR13.54, 95 CI 6.66–27.51, P = 0.001) male gender age (aOR 4.73, 95 CI 2.91–7.68, P = 0.001), low BMI (aOR 4.48, 95 CI 3.26–6.15; P = 0.001), diabetes (aOR 1.67, CI 0.99–2.83, P = 0.055), previous history of TB (aOR3.64, 95 CI 2.20–6.02, P = 0.001), and alcohol users (aOR 1.62, 95 CI 1.14–2.32, P = 0.007) were associated with an increased risk of developing PTB (Fig 2).

## Discussion

The salient finding from this community survey that the prevalence of bacteriologically positive PTB was 307 per 100000 population. It was found to be varied in different socio-economic groups from 46 in 15–34 years of age group to 2778 per 100000 population those who had a previous history of TB. It was corroborated with the surveys conducted in other parts of the country which also reported that there is a regional variation of PTB prevalence that ranged from 171 to 528 per 100000 population and the pooled estimate of sub-national TB prevalence was 300 per 100000 [6]. Our study also found that almost similar prevalence to these estimates (300 vs 307).

However, this was higher as compared to other parts of the country Bangalore rural Karnataka [7], SAS Nagar (Punjab) [8], Wardha (Maharastra) [9], and Faridabad (Haryana) [10]. The prevalence was significantly less as compared to Gujarat (439.2) [11], Kanpur (481.1) [7], Banda (528.4) [7] and Jabalpur (337) [12]. This survey was done in rural areas of Thiruvallur District while comparing this prevalence of PTB within Tamil Nadu state it was significantly less as compared to urban Chennai (307 vs 357) [13].

With respect to community based PTB prevalence survey conducted in the same area, it was found that prevalence of smear-positive (130 vs 326, 259, 168, 180) and culture positive PTB (277 vs 607, 454, 309, 388) was significantly less in the current survey (2015–18) as compared to previous surveys done from 1999 to 2008 at two and half years intervals [14–16]. It revealed that there was a reduction in smear-positive PTB by 60% reduction from 326 to 130 and reduction in culture positive PTB was 54.4% reduction during the 20 years' time. It may be progression owing to the government's intensive of TB control activities such as the implementation of Directory Observed Treatment Short-course, implementation of new diagnostic tools for early detection, repeated community survey to do active case finding and IEC programmes.

This study has established that increasing age, male gender, low BMI, diabetes, history of TB and alcohol consumption were the risk of developing PTB. These are the known risk factors that were reported from various studies [17–21]. We rephrasing that there is a need to take appropriate action to modify among these modifiable risk factors, so that PTB reduction can be accelerated.

Prediction upon outcomes of this current study with the rational limitation that it was carried out at Thiruvallur district of the southern part of Tamil Nadu. It is required to contact the prevalence survey of PTB in other parts of the country for better understanding. The significance of this current study was that ICMR-NIRT had done repeated community-based PTB periodic prevalence surveys in the same area. This is a unique opportunity to find out the impact of the National TB Control Programme.

## Conclusion

The current study, estimated community-based prevalence of bacteriologically confirmed PTB and its risk factors. This finding highlights that the study area continues to be a high TB burden area indicating the prevalence of 307 bacteriologically positive TB cases per 100000 population. The prevalence of PTB was significantly higher among the elderly population, males, those who have low BMI, diabetes, those who have had a previous history of TB and alcohol users. In addition, our finding demonstrates that there is a reduction in the prevalence of PTB in this area attributing for better implementation and success of the National TB Control Programme in this district. Our study suggests that there is a need to introduce risk reduction interventions to accelerate the reduction of PTB. Further, there is a need to conduct surveys to evaluate the sustainability of positive decline trend.

## Acknowledgments

The authors are thankful to the Central TB Division, Ministry of Health and Family Welfare (MoHFW), Government of India and State TB Division, MoHFW, Government of Tamil Nadu given permission, and their overall coordination to undertake this TB prevalence survey. We acknowledge the Scientific Advisory Committee and Institutional Ethics Committee for their approval and their suggestions to improve the study. We are grateful to all Epidemiology field survey staff, Electronic Data Management Unit, Bacteriology laboratory for their support. We wish to express our deep gratitude to all participants in this survey.

## Author Contributions

**Conceptualization:** Chandra Kumar Dolla, Bhaskaran Dhanaraj, Sriram Selvaraj, Rajesh Mondal, Srikanth Prasad Tripathy.

**Data curation:** Kannan Thiruvengadam, Rajendran Krishnan, Muniyandi Malaisamy, Lakshmi Murali.

**Formal analysis:** Kannan Thiruvengadam, Rajendran Krishnan, Muniyandi Malaisamy, Srinivasa B. Marinaik.

**Funding acquisition:** Chandra Kumar Dolla, Bhaskaran Dhanaraj, Padmapriyadarsini Chandrasekaran, Sriram Selvaraj, Srikanth Prasad Tripathy.

**Investigation:** Pradeep Aravindan Menon, Rajesh Mondal.

**Methodology:** Chandra Kumar Dolla, Bhaskaran Dhanaraj, Padmapriyadarsini Chandrasekaran, Sriram Selvaraj, Pradeep Aravindan Menon, Kannan Thiruvengadam, Rajendran Krishnan, Muniyandi Malaisamy, Lakshmi Murali.

**Project administration:** Rajesh Mondal, Srikanth Prasad Tripathy.

**Software:** Pradeep Aravindan Menon, Kannan Thiruvengadam, Rajendran Krishnan, Srinivasa B. Marinaik.

**Supervision:** Chandra Kumar Dolla, Bhaskaran Dhanaraj, Padmapriyadarsini Chandrasekaran, Sriram Selvaraj, Pradeep Aravindan Menon, Srinivasa B. Marinaik, Lakshmi Murali, Srikanth Prasad Tripathy.

**Validation:** Pradeep Aravindan Menon, Rajesh Mondal.

**Writing – original draft:** Chandra Kumar Dolla, Padmapriyadarsini Chandrasekaran.

**Writing – review & editing:** Bhaskaran Dhanaraj, Sriram Selvaraj, Pradeep Aravindan Menon, Kannan Thiruvengadam, Rajendran Krishnan, Rajesh Mondal, Muniyandi Malaisamy, Srinivasa B. Marinaik, Lakshmi Murali, Srikanth Prasad Tripathy.

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
