## [Decision Letter · Decision Letter 0]

6 Aug 2020

PONE-D-20-16942

Prevalence of bacteriologically confirmed pulmonary tuberculosis and associated risk factors: A community survey in Thirvallur District, south India

PLOS ONE

Dear Dr. Kumar,

Thank you for submitting your manuscript to PLOS ONE. After careful consideration, we feel that it has merit but does not fully meet PLOS ONE’s publication criteria as it currently stands. Therefore, we invite you to submit a revised version of the manuscript that addresses the points raised during the review process.

Please submit your revised manuscript. If you will need more time than this to complete your revisions, please reply to this message or contact the journal office at plosone@plos.org. Please include the following items when submitting your revised manuscript:

We look forward to receiving your revised manuscript.

Kind regards,

Frederick Quinn

Academic Editor

PLOS ONE

Journal Requirements:

2.We suggest you thoroughly copyedit your manuscript for language usage, spelling, and grammar. If you do not know anyone who can help you do this, you may wish to consider employing a professional scientific editing service.  

4.Thank you for stating the following in the Acknowledgments Section of your manuscript:

 [We are grateful for the financial and logistical support from the World Health Organisation, India

and Indian Council of Medical Research, Department of Health Research, MoHFW, New

Delhi.]

 [The funders had no role in study design, data collection and analysis, decision to publish, or preparation of the manuscript.]

Reviewers' comments:

Reviewer's Responses to Questions

**Comments to the Author**

1. Is the manuscript technically sound, and do the data support the conclusions?

Reviewer #1: Yes

2. Has the statistical analysis been performed appropriately and rigorously? 

Reviewer #1: I Don't Know

3. Have the authors made all data underlying the findings in their manuscript fully available?

Reviewer #1: Yes

4. Is the manuscript presented in an intelligible fashion and written in standard English?

Reviewer #1: No

5. Review Comments to the Author

Reviewer #1: Review of ms # PONE-D-20-16942 submitted to PLOS ONE by Kumar et. Al.

Tuberculosis remains one of the major respiratory diseases in southeast asia including the Indian subcontinent and a variety of environmental factors and co-morbidities determine not only the pathology of TB but also success of TB treatment. In this manuscript, Kumar et. al provide a summary of active TB surveillance done in the south Indian Thiruvallur district. Community based surveillance targeting ~69,000 individuals >15y of age included screening for pulmonary TB using sputum samples and chest X-rays. The authors also collected demographical information as well as smoking, alcohol consumption and established correlation with a positive TB outcome. Their data shows a decreasing trend of TB prevalence in this sample set supporting the efficacy of the national strategic plan for TB containment. The study conclusions support previous data on TB incidence and correlation with comorbidities and behavioural traits (tobacco and /or alcohol consumption). Thus, while not novel, the data show the efficacy of the TB NSP and underscore the need for continuing TB surveillance and research. The major critique of the manuscript is there are a lot of typographical errors in the manuscript and these need to be corrected to improve readership.

6. PLOS authors have the option to publish the peer review history of their article (what does this mean?). If published, this will include your full peer review and any attached files.

Reviewer #1: No

---

## [Author Response · Author response to Decision Letter 0]

5 Oct 2020

Reviewers comments 

The manuscript must describe a technically sound piece of scientific research with data that supports the conclusions. Experiments must have been conducted rigorously, with appropriate controls, replication, and sample sizes. The conclusions must be drawn appropriately based on the data presented. Tuberculosis remains one of the major respiratory diseases in south East Asia including the Indian subcontinent and a variety of environmental factors and co-morbidities determine not only the pathology of TB but also success of TB treatment. In this manuscript, Kumar et. al provide a summary of active TB surveillance done in the south Indian Thiruvallur district. Community based surveillance targeting ~69,000 individuals >15y of age included screening for pulmonary TB using sputum samples and chest X-rays. The authors also collected demographical information as well as smoking, alcohol consumption and established correlation with a positive TB outcome. Their data shows a decreasing trend of TB prevalence in this sample set supporting the efficacy of the national strategic plan for TB containment. The study conclusions support previous data on TB incidence and correlation with comorbidities and behavioural traits (tobacco and /or alcohol consumption). Thus, while not novel, the data show the efficacy of the TB NSP and underscore the need for continuing TB surveillance and research. The major critique of the manuscript is there are a lot of typographical errors in the manuscript and these need to be corrected to improve readership.

Reply to Reviewers comments 

Thank you for the appreciation on our work and comments. We are submitting the revised version of the manuscript entitled “Prevalence of bacteriologically confirmed pulmonary tuberculosis and associated risk factors: A community survey in Thirvallur District, south India”. As suggested by reviewer we carefully checked and English corrections made in the manuscript with track changes. I hope this revised version of the manuscript is most suitable for publication. I am very happy to provide any more details and further clarifications.

---

## [Decision Letter · Decision Letter 1]

4 Jan 2021

PONE-D-20-16942R1

Prevalence of bacteriologically confirmed pulmonary tuberculosis and associated risk factors: A community survey in Thirvallur District, south India

PLOS ONE

Dear Dr. Kumar,

Thank you for submitting your manuscript to PLOS ONE. After careful consideration, we feel that it has merit but does not fully meet PLOS ONE’s publication criteria as it currently stands. Therefore, we invite you to submit a revised version of the manuscript that addresses the points raised during the review process.

Please submit your revised manuscript. If you will need significantly more time to complete your revisions, please reply to this message or contact the journal office at plosone@plos.org. Please include the following items when submitting your revised manuscript:

We look forward to receiving your revised manuscript.

Kind regards,

Frederick Quinn

Academic Editor

PLOS ONE

Reviewers' comments:

Reviewer's Responses to Questions

**Comments to the Author**

1. If the authors have adequately addressed your comments raised in a previous round of review and you feel that this manuscript is now acceptable for publication, you may indicate that here to bypass the “Comments to the Author” section, enter your conflict of interest statement in the “Confidential to Editor” section, and submit your "Accept" recommendation.

Reviewer #2: (No Response)

2. Is the manuscript technically sound, and do the data support the conclusions?

Reviewer #2: Partly

3. Has the statistical analysis been performed appropriately and rigorously? 

Reviewer #2: Yes

4. Have the authors made all data underlying the findings in their manuscript fully available?

Reviewer #2: Yes

5. Is the manuscript presented in an intelligible fashion and written in standard English?

Reviewer #2: Yes

6. Review Comments to the Author

Reviewer #2: In this manuscript, Dolla et al. summarized an epidemiology study on active pulmonary tuberculosis (PTB) at Tiruvallure district via cluster sampling method. The authors defined PTB according to positive smear test or bacterial culture test. Out of 62494 screened population, the authors determined a PTB prevalence rate as 307 per million population. The authors further reported positive correlation between PTB with decreased BMI, smoking, diabetes and elder population. The study overall is convincing, however the authors need to address the following issues:

(1) The BCG vaccination does not have protection judged by a p value of 0.792. Can the authors comment on the possible reasons?

(2) The authors applied p value test to justify the statistical significance between 2 groups. However, the authors need to specify the mathematical method and type-II error for statistical power design, which they leveraged to justify their statistics with a smaller sample size compared to the original one.

7. PLOS authors have the option to publish the peer review history of their article (what does this mean?). If published, this will include your full peer review and any attached files.

Reviewer #2: No

---

## [Author Response · Author response to Decision Letter 1]

15 Jan 2021

Thank you for this valid question. We would like to clarify that it was documented that the protective efficacy of BCG was effective for children protecting from the severe forms of TB. 

Reference: Subramani R, Datta M, Swaminathan S. Does effect of BCG vaccine decrease with time since vaccination and increase tuberculin skin test reaction? Indian J Tuberc 2015; 62(4): 226-9. 

The authors applied p value test to justify the statistical significance between 2 groups. However, the authors need to specify the mathematical method and type-II error for statistical power design, which they leveraged to justify their statistics with a smaller sample size compared to the original one.

We used a univariate and multivariate logistic regression model to look for any association between TB and risk factors such as age, gender, BMI, diabetes, hypertension, history of TB tobacco smoking, alcohol consumption and cooking fuel used in the household. 

We made use of the algorithm suggested by Van Wijngaarden-Dekker-Brent algorithm (Brent, 1973) to test the statistical power of the regression models which are more than 80% as required to ensure the statistical significance.

---

## [Editor Report · Decision Letter 2]

4 Feb 2021

Prevalence of bacteriologically confirmed pulmonary tuberculosis and associated risk factors: A community survey in Thirvallur District, south India

PONE-D-20-16942R2

Dear Dr. Kumar,

We’re pleased to inform you that your manuscript has been judged scientifically suitable for publication and will be formally accepted for publication once it meets all outstanding technical requirements.

Kind regards,

Frederick Quinn

Academic Editor

PLOS ONE
---

## [Editor Report · Acceptance letter]

24 Sep 2021

PONE-D-20-16942R2 

Prevalence of bacteriologically confirmed pulmonary tuberculosis and associated risk factors: A community survey in Thirvallur District, south India 

Dear Dr. Dolla:

I'm pleased to inform you that your manuscript has been deemed suitable for publication in PLOS ONE. Congratulations! Your manuscript is now with our production department. 

Kind regards, 

on behalf of

Dr. Frederick Quinn 

Academic Editor

PLOS ONE